# The Endocannabinoid System and Physical Activity—A Robust Duo in the Novel Therapeutic Approach against Metabolic Disorders

**DOI:** 10.3390/ijms23063083

**Published:** 2022-03-12

**Authors:** Tomasz Charytoniuk, Hubert Zywno, Klaudia Berk, Wiktor Bzdega, Adrian Kolakowski, Adrian Chabowski, Karolina Konstantynowicz-Nowicka

**Affiliations:** 1Department of Physiology, Medical University of Bialystok, 15-089 Bialystok, Poland; tomasz.charytoniuk@umb.edu.pl (T.C.); klaudia.berk@umb.edu.pl (K.B.); wbzdega@gmail.com (W.B.); adriankolakowski17@gmail.com (A.K.); adrian@umb.edu.pl (A.C.); karolina.konstantynowicz@umb.edu.pl (K.K.-N.); 2Department of Ophthalmology, Antoni Jurasz University Hospital No. 1, 85-094 Bydgoszcz, Poland

**Keywords:** endocannabinoid system, physical activity, obesity, cannabinoid receptor, anandamide, 2-arachidonylglycerol

## Abstract

Rapidly increasing worldwide prevalence of obesity and related pathologies encompassing coronary heart disease, hypertension, metabolic syndrome, or type 2 diabetes constitute serious threats to global health and are associated with a significantly elevated risk of premature death. Considering the enormous burden of these pathologies, novel therapeutic and preventive patterns are indispensable. Dysregulation of one of the most complex biological systems in the human body namely, the endocannabinoid system (ECS) may result in metabolic imbalance and development of insulin resistance, type 2 diabetes, or non-alcoholic fatty liver disease. Furthermore, many studies showed that physical exercises, depending on their type, intensity, and frequency, exert various alterations within the ECS. Emerging evidence suggests that targeting the ECS via physical activity may produce robust beneficial effects on the course of metabolic pathologies. However, the data showing a direct correlation between the ECS and physical activity in the aspect of metabolic health are very scarce. Therefore, the aim of this review was to provide the most up-to-date state of knowledge about the interplay between the ECS activity and physical exercises in the novel therapeutic and preventive approach toward metabolic pathologies. We believe that this paper, at least in part, will fulfill the existing gap in knowledge and encourage researchers to further explore this very complex yet interesting link between the ECS, its action in physical activity, and subsequent positive outcomes for metabolic health.

## 1. Introduction

Nowadays, metabolic disorders, e.g., obesity, metabolic syndrome, or type 2 diabetes mellitus (T2DM) are some of the major medical concerns that occur among different populations, especially those with a predominance of the Western dietary pattern [1,2]. It is estimated that about 2 billion people across the world are overweight or obese. Consequently, obesity and its associated pathologies constitute serious threats to global health and are widely regarded as a worldwide epidemic, thereby posing as one of the leading causes of death in well-developed countries [3]. Various and complex molecular mechanisms are involved in the pathogenesis of these disorders, including insulin resistance, chronic inflammation, defective metabolism within adipose tissue, excessive oxidative stress, and alterations in the endocannabinoid system (ECS) [4,5]. The ECS constitutes a versatile and complex regulator of human energy homeostasis, which is involved in metabolic health functions, for instance, appetite, food intake, energy expenditure, metabolism of carbohydrates and lipids, likewise hedonic rewards such as palatability [6,7]. Recent studies indicate that dysregulation in the components of ECS is linked with the metabolic imbalance and, subsequently, the development of obesity and related disorders. Interestingly, the ECS is widely known for its interplay with physical activity (PA) [7,8]. Although the correlation between the ECS and PA was described in the literature, underlying molecular alterations remain unknown and deserve to be reviewed more extensively. A body of evidence suggests that targeting the ECS signaling via exercise may result in significant beneficial outcomes either in the supportive treatment or prophylaxis of metabolic disorders [9,10]. Therefore, considering the scarcity of studies showing this relationship, this review aimed to provide the most up to date state of knowledge about the interplay between the ECS and PA in the novel therapeutic and preventive approach towards metabolic pathologies, such as obesity, insulin resistance, type 2 diabetes mellitus, and non-alcoholic fatty liver disease (NAFLD).

## 2. The Endocannabinoid System—A Brief Overview of the Key Functions and Elements

The ECS constitutes one of the most complex biological systems in the human body, creating a milieu responsible for the plethora of essential functions, including maintaining metabolic and cardiovascular homeostasis, mediating immune responses, and modulating the signaling within reward systems, as well as playing a relevant role in the brain physiology components, such as mood, cognition, and neurogenesis [11,12,13]. Furthermore, the ECS is involved in fertility, reproduction, and pregnancy [14]. In general, the ECS comprises (1) cannabinoid receptors (CBRs), (2) endocannabinoids (eCBs)—lipid ligands of cannabinoid receptors originating from omega-3 and omega-6 polyunsaturated fatty acids, among which anandamide (N-arachidonoyl-ethanolamine, AEA) and 2-arachidonoylglycerol (2-AG) received the most attention and were best described in the literature, (3) enzymes involved in bioformation and degradation of eCBs [15,16]. Cannabinoid receptors type 1 and type 2 (CB1R and CB2R) belong to the family of 7-transmembrane G-protein-coupled receptors (GPCRs) (Gi/o); thus, their activation, by suitable ligand, leads to the inhibition of adenyl cyclase and decreases in the concentration of cyclic adenosine monophosphate (cAMP). As a result, cAMP intensifies the activity of p42/p44 mitogen-activated protein kinase (MAPK) and Jun N-terminal kinase (JNK) which activate divergent nuclear transcription factors and alter the cellular metabolism. Additionally, cannabinoid receptors activate potassium and inhibit calcium channels in the presynaptic end of neurons that results in the overall inhibition of neurotransmitter release [17,18,19]. The distribution of CB1R is mainly clustered within brain structures, especially the neocortex, hippocampus, basal ganglia, and brain stem; likewise, its presence was confirmed in the skeletal muscles, lungs, testes, gastrointestinal tract, liver, pancreas, and adipose tissue. In turn, CB2R is mainly expressed on the immune cells (macrophages, T lymphocytes, B lymphocytes), hematopoietic stem, and progenitor cells present in bone marrow, as well as in the thymus and lymph nodes; thereby its activation triggers wide-range of immunomodulatory effects (Figure 1) [19,20,21,22]. Endocannabinoids are synthesized on demand in response to the elevated intracellular Ca^2+^ concentration [18]. 2-AG, which is synthesized from diacylglycerol (DAG), acts as a full agonist with a moderate affinity of cannabinoid receptors, whereas is hydrolyzed by monoacylglycerol lipase (MAGL) [23,24]. In turn, AEA constitutes a partial agonist of cannabinoid receptors synthesized from N-arachidonoyl phosphatidylethanolamine (NAPE) coordinated by N-acyl-phosphatidylethanolamine phospholipase D-like esterase (NAPE-PLD) and is degraded to arachidonic acid (AA) by the activity of fatty acid amid hydrolase (FAAH) [15,18]. Next to the well-known eCBs, analogous “endocannabinoid-like” lipid ligands were described namely oleoylethanolamine (OEA) and palmitoylethanolamide (PEA). Although OEA and PEA do not bind typical cannabinoid receptors, they possess a high affinity toward peroxisome proliferator-activated receptor alpha (PPARα) and orphan GPCRs. These compounds play an important role in hypoxia-induced intestinal permeability and suppression of the inflammation within the gastrointestinal tract, which highlights their potential importance in the aspect of cancer research [25,26]. Recently some researchers provided the term “endocannabinoidome” (eCBome), which refers to the broad and expanded signaling system that, apart from the aforementioned elements, contains a large number of different membrane and nuclear receptors, i.e., PPARs, transient receptor potential vanniloid 1 ion channel (TRPV1), various GPCRs, their endogenous ligands, and dozens of enzymes playing important roles in endocannabinoid signaling. Endocannabinoidome may constitute a promising target in the therapy of various pathologies in which the components of ECS seem to be involved; therefore, further research on the emerging new metabolic implications of eCBome remains in high demand [16,17,27,28].

## 3. The Endocannabinoid System and Metabolic Pathologies

Many researchers indicated that ECS might be widely involved in the regulation of energy balance and feeding behavior through both central and peripheral mechanisms [27]. Central mechanisms include hedonic energy regulation in the nucleus accumbens (NAc) and the ventral tegmental area that results in a higher motivation via the reward system. Moreover, another essential mechanism that leads to an increased appetite and higher food intake is homeostatic regulation in the hypothalamus, a major site of ECS action. Conversely, in the peripheral levels of action, ECS is widely associated with reduced expression of adiponectin, a protein hormone involved in suppressing glucose production in the liver and enhancing fatty acid oxidation in skeletal muscles. Another mechanism of ECS action is decreased AMPK activity (5’AMP-activated protein kinase), which is involved in insulin sensitivity, and its inhibition regulates lipogenesis and fatty acids or cholesterol synthesis in both the liver and adipose tissue [28]. Interestingly, the concentration of anandamide might be decreased via leptin action (an appetite-reducing adipose tissue-origin hormone that plays an essential role in food intake, body weight control, and metabolism), which enhances FAAH expression and leads to a lower concentration of anandamide in adipose tissue [29,30]. It is commonly known that low circulating levels of the leptin, as well as leptin resistance, are widely correlated with a higher risk of obesity and its aftermath [31,32]. Interestingly, a recent study conducted by Tam et al. indicated that a peripheral inhibition of CB1R in mice with diet-induced obesity (DIO) leads to an increase in the leptin sensitivity, which results in hypophagia via the reactivation of melanocortin signaling in the arcuate nucleus (ARC) in the hypothalamus [33]. Some studies reported that anandamide may lead to increased liver lipogenesis via the CB1R [34,35]. Another study by Karaliota et al. demonstrated that AEA may act as a PPAR-γ agonist and subsequently leads to the development of ECS-related adipogenesis in primary rat preadipocytes isolated from epididymal adipose tissue of male Wistar rats [36]. Numerous studies have indicated that endocannabinoids widely affect proliferation, as well as differentiation of adipocytes [33,37,38]. Amidst various components of endocannabinoidome, researchers point out the relevance of the link between the ECS and ion channel TRPV1 in ensuring undisturbed energy homeostasis. It is highly expressed within the central nervous system as well as peripheral tissues, including stomach and adipose tissue. TRPV1 is a non-selective cation channel activated by capsaicin, high temperature, and acidosis that belong to the transient receptor potential protein superfamily [39]. Moreover, among various stimuli which interact with TRPV1, N-acyldopamines, leukotriene B4 and N-acylethanoloamines (e.g., AEA) may be distinguished as TRPV1 agonists. The effect of endocannabinoids on TRPV1 may be exerted through direct activation on this channel or indirectly through activation of the CB1R that activates the phospholipase C-PKC pathway. However, endocannabinoids may also activate CB1R, which leads to the inhibition of the adenylate cyclase-PKA pathway, resulting in diminished TRPV1 activity. Those correlations are still unclear and, at least in part, depend on the concentration of AEA [37,40]. The metabolic functions of TRPV1 include its influence on glucose metabolism and lipid oxidation, regulation of the appetite (via interaction with appetite-regulating hormones e.g., ghrelin and leptin) as well as activation of thermogenesis [40]. Numerous studies highlighted that impaired signaling within TRPV1 may contribute to the development of various metabolic disorders, including obesity and T2DM [38,41]. Therefore, the complex interactions between the ECS and TRPV1 are still a subject of discussion and should be clarified in future studies. 

### 3.1. Obesity and Non-Alcoholic Fatty Liver Disease (NAFLD)

A substantial correlation between metabolic disturbances and endocannabinoid system overactivation was unequivocally demonstrated in both in vivo animal studies and clinical trials [42,43]. So far, a vast number of studies have linked obesity with higher circulating levels of endocannabinoids [44,45]. It is known that obesity and higher food intake might lead to an overactivation of ECS by increased cannabinoid receptor activity. Moreover, this might become a vicious circle—obesity-related ECS activation might contribute to an intensified lipogenesis, higher food intake, and consequently lead to further fat storage. Increased fat accumulation in adipose tissue might be associated with activation of CB1R since it was indicated that those receptors might increase the activity of lipoprotein lipase, a key enzyme regulating triglyceride hydrolysis to free fatty acids, which are subsequently transported and deposited in the liver [46]. Thus, inhibition of CB receptors seems to be relevant.

Interestingly, rimonabant was the first CB1R antagonist approved for the treatment of obesity since clinical trials revealed that it might promote weight loss in obese patients. However, due to a number of serious psychiatric adverse effects, e.g., anxiety or depression, the clinical use of rimonabant was discontinued, and the drug was withdrawn from the worldwide market [47]. A human-based study by Bennetzen et al. demonstrated that levels of 2-arachidonoylglycerol were significantly reduced in the abdominal adipose tissue, whereas weight loss led to a normalization of this endocannabinoid. Furthermore, the same study indicated that the low expression of CB1 in abdominal adipose tissue was normalized after weight loss, whereas in gluteal adipose tissue the CB1 expression was decreased after weight loss. Thus, this research, together with other related studies, presented substantial results that ECS is widely dysregulated among individuals suffering from obesity [48,49] Moreover, it indicates that not only CB receptor expression is affected by obesity but also levels of ECs. Indeed, some researchers reported that in obese patients, both AEA and 2-AG levels are increased, and elevated plasma concentrations of endocannabinoids (ECs) might be correlated with obesity aftermath, e.g., cardiovascular diseases, as well as non-alcoholic fatty liver disease (NAFLD) [50,51,52]. Moreover, increased concentrations of endocannabinoids observed in small intestinal mucosa and plasma of high-fat diet-fed obese mice inhibited may change TRPV1 activity, which is involved in energy homeostasis [53]. However, because of various interactions of this channel with other systems, the precise role of TRPV1 in regulating of food intake and energy expenditure needs further investigation. All these outcomes highlight those alterations in dysregulated ECS signaling are very complex and should be investigated in depth. 

Non-alcoholic fatty liver disease is an obesity-related metabolic pathology induced by enhanced triacylglycerols accumulation leading to hepatic steatosis, excluding alcohol abuse as a pathogenic factor [54]. Numerous studies have indicated that NAFLD occurrence is widely correlated with the dysfunction of the endocannabinoid system because lipid accumulation in the liver might be mediated via alterations in ECS [55,56]. It is known that the basal hepatic expression of cannabinoid receptors is indistinct, with low levels of both CB1R and CB2R. However, an increased expression of the cannabinoid receptors was proven in NAFLD. Moreover, CB1R was shown as a key mediator of insulin resistance development, increased liver lipogenesis, and steatosis in both animal and human studies, whereas CB2R might be considered as a promoter of inflammatory processes underlying the pathogenesis of liver steatosis [57]. It is known that blockage of cannabinoid receptor type 1 may result in attenuation of hepatic oxidative stress, as well as the impairment of inflammatory response by inhibiting the production of pro-inflammatory cytokines [58,59] Furthermore, Irungbam et al. recently demonstrated in both in vitro (AML12 liver cell line) and in vivo studies on mice with a global CB1R receptor knockout that lower CB1R signaling reduced liver steatosis via downregulation of perilipin 2—the protein that is involved in the suppression of lipolysis and lipid accumulation [60]. Shi et al. conducted in vitro studies in which application of CB1R antagonist in HepG2 cells resulted in a significant decrease in factors widely involved in lipogenesis, i.e., sterol regulatory element-binding protein (SREBP1c), carbohydrate responsive element-binding protein (ChREBP), liver X receptors (LXRs), acetyl-CoA carboxylase (ACC1) and fatty acid synthase (FAS) [61]. Moreover, Deveaux et al. conducted a study on mice which indicated that the inactivation of cannabinoid receptor type 2 reduced steatosis, liver triacylglycerol concentration, and obesity-associated inflammation caused by a high-fat diet [62]. Interestingly, CB1R was defined as a pro-fibrogenic activator, while cannabinoid receptor type 2 was proposed to have anti-fibrogenic properties in the liver [35,63]. Thus, ECs and their receptors might be considered as a possible and promising target for the treatment of liver steatosis and its metabolic aftermath.

### 3.2. Insulin Resistance and Type 2 Diabetes

Insulin resistance (IR), defined as a condition with impaired insulin action and response in tissues, is widely considered to be one of the most fundamental factors associated with the development of many metabolic disorders, e.g., type 2 diabetes mellitus or metabolic syndrome [64,65]. A vast number of studies has provided evidence indicating that activation of peripheral CB1R located in insulin-dependent tissues, i.e., liver, adipose tissue, or skeletal muscles, is widely associated with the development of dyslipidemia, insulin resistance, and type 2 diabetes [66]. Additionally, plenty of studies reported that CB1R blockage leads to increased insulin sensitivity and, subsequently, improves both glucose and lipid metabolism [67,68]. Considering the pathophysiological basis of IR development, Liu et al. demonstrated that it is correlated with activation of CB1R that inhibits insulin signaling via endoplasmic reticulum stress-dependent suppression of enzymes involved in the insulin signaling pathway, e.g., PKB/Akt [69]. Interestingly, further Liu et al. studies provided notable results that blockage of CB1R in obese mice improved glycemic control through the hepatic Sirt1/mTORC2/Akt pathway, as well as increased fatty acid β-oxidation via LKB1/AMPK signaling route [70]. A study conducted by Jourdan et al. demonstrated that intravenous administration of β-D-glucan-encapsulated siRNA to knock down CB1R gene expression, improved insulin sensitivity in Kupfer cells of mice with obesity. Thus, it might be said that the CB1R played a significant role in the development of liver insulin resistance via Kupffer cell-related inflammatory mechanisms [71]. Considering the CB2R, principally expressed in immune tissues, several in vitro and in vivo studies demonstrated CB2R-mediated anti-inflammatory effects of endocannabinoids, i.e., AEA and 2-AG, as well as synthetic CB2R agonists in models of various metabolic diseases, e.g., obesity [72,73,74]. Furthermore, Verty et al. reported that activation of CB2R reduced the inflammatory response and promoted anti-obesity effects by reducing body weight, while not changing mood-related behaviors [75]. Although plenty of studies support the claim that endocannabinoids are anti-inflammatory mediators, a limited amount of evidence demonstrated the pro-inflammatory action of endocannabinoids via CB2R activation [76,77]. Moreover, Turcotte et al. showed that most of the pro-inflammatory effects involve 2-AG, but not AEA [78].

Considering the correlation between the endocannabinoid system and oxidative stress, a study conducted by Mendizabal-Zubiaga et al. showed that the mitochondrial activation of CB1R in muscle cells might be correlated with the mitochondrial regulation of oxidative activity. Thus, this might be highly associated with the development of insulin resistance, since mitochondrial dysfunction is correlated with a loss of muscular oxidative capacity in response to a high intake of fatty acids, and therefore may be involved in the development of metabolic disorders [79,80]. Notably, Dipanjan et al. demonstrated that activation of the CB1R is involved in gluconeogenesis via direct activation of an endoplasmic reticulum (ER) membrane-localized stress-dependent liver-specific transcription factor called CREBH—cyclic AMP-responsive element-binding protein H, which is considered to be one of the key mediators in both glucose and lipid metabolism [81,82]. Thus, it may be said that there is a molecular association between the activation of the endocannabinoid system and levels of circulating glucose.

An endocannabinoid-related study conducted by Jourdan et al. on the Zucker diabetic rats indicated that the loss of pancreatic beta cell functions and its amount is associated with the release of pro-inflammatory cytokines, i.e., IL-1β and IL-18. Those molecules, acting as paracrine substances that induce beta-cell apoptosis, are released from infiltrating M1 macrophages as a result of macrophages-origin CB1R signaling that participates in selective activation of the Nlrp3-ASC inflammasome [83]. Moreover, a recent study by Kim et al. conducted on pancreatic beta-cell lines MIN6 and βTC6 incubated with synthetic CB1R agonist WIN55,212-2 showed a decreased expression of anti-apoptotic protein Bcl-2 and cell cycle regulator cyclin D2, together with caspase-3-dependent apoptosis afterward [84]. Thus, these two studies unequivocally determine the effect of CB1R activation in beta-cell apoptosis and therefore insulin signaling deficiency and the development of type 2 diabetes. It is commonly known that insulin resistance, as well as T2DM, are widely associated with an increased tissue concentration of sphingolipids—a class of lipids whose fractions are also involved in cellular growth, differentiation, and regulation of apoptosis in addition to playing an essential role as structural molecules in cellular membranes [85,86]. Cinar et al. linked the action of the endocannabinoid system and insulin resistance development together with excessive sphingolipids deposition. It was indicated in a study conducted on C57Bl6/J mice with high-fat diet-induced obesity (DIO) that the factor, which is widely involved in the development of hepatic IR, is increased de novo ceramides synthesis, which is mediated by endoplasmic reticulum (ER) stress-dependent activation of CB1R [87]. Thus, it may be assumed that changes in CB receptors and other components of the ECS may be associated with a favorable prognosis not only of obesity and NAFLD but also IR and type 2 diabetes.

## 4. The Endocannabinoid System and Physical Activity 

Physical activity is broadly portrayed as one of the most popular, potent, cost-friendly, and pleasant ways to maintain a prolonged and healthy life. It constitutes a major pillar in the treatment of metabolic disorders next to the proper diet, pharmacotherapy, chemoprevention, and in the most severe cases, bariatric interventions [88]. What is more, it affects human mental health, leading to mood, perception, concentration, and creativity improvement as well as producing a very interesting phenomenon called “runner’s high” [89]. Despite widely known positive effects in the therapy of various diseases, the exact molecular mechanisms induced by PA are still a subject of discussion. Interestingly, many studies outlined the influence of different types of physical activity on endocannabinoid signaling. These results, however, seem to be partially unclear and confusing; therefore, they deserve to be gathered and described comprehensively. The study conducted by Sparling et al. was arguably the first showing that moderate exercises led to the increased level of AEA but not 2-AG in the sera of healthy subjects after 50 min of treadmill running or cycling on a stationary bike at the level of 70–80% of maximum heart rate [90]. Other studies also indicated that PA, especially at medium intensity, led to the increased blood level of AEA, whereas 2-AG remained unchanged [91,92,93,94]. Heyman et al. revealed the link between significantly increased plasma AEA, OEA, and PEA levels during intense exercise followed by 15 min of recovery and elevated serum brain-derived neurotrophic factor (BDNF) levels, thereby indicating an important role of the ECS in the production of neuroplastic and antidepressant effects triggered by PA. A simultaneous increase in OEA and PEA along with AEA probably comes from the fact that these compounds share similar synthesis pathways, whereas 2-AG formation seems to be regulated by different metabolic routes [95]. Conversely, the study conducted by Cedernaes et al. showed elevated blood 2-AG and OEA levels after acute ergometer exercises, while AEA remained stable [96]. The possible reason for this discrepancy may lie in the fasted status of the participants before activity [97]. Brellenthin et al. observed that both AEA and 2-AG are increased after aerobic exercises in the blood of healthy individuals, but the increase in AEA was more substantial in the prescribed by the investigator activity in comparison with self-selected ones, based on personal preferences [98]. Thompson et al. in their study on mice showed that males after exercise tended to have more increased 2-AG serum levels, whereas AEA content was higher in female subjects. These results raised an interesting concern that alterations in the circulating eCBs induced by PA may be at least partially, sex-dependent; however, more data is needed to clarify this issue [99]. It may be suspected that various changes in the levels of eCBs may be a result of different type and time of exercise, but they all exerted changes in the nervous system. Furthermore, Crombie et al. showed that acute isometric activity led to the elevated plasma levels of both AEA and 2-AG as well as OEA and PEA, which resulted in overall analgesic effects [100]. On the other hand, most animal studies pointed out that aerobic activity, i.e., swimming or treadmill running, was associated with increased expression and density of both CB1R and CB2R within the central nervous system, namely the hippocampus, striatum, and spinal cord, that were associated with positive neurological outcomes, including improved cognition, analgesia, and reduced neuroinflammation [101,102,103]. Nevertheless, there is evidence, that PA may lead to the downregulated expression of cannabinoid receptors in the aforementioned structures which suggests that further investigation is required to elucidate these discrepancies [101,104]. Physical activity, depending on such factors as type, frequency, intensity, and duration leads to the (I) activation of the ECS signaling, (II) significantly affects circulating blood levels of the endocannabinoids, mainly AEA and 2-AG, as well as (III) change in expression of the cannabinoid receptors (CB1R and CB2R). Despite the fact, that molecular interaction between the ECS and PA has been well described in the literature, it is still difficult to point out clear, unequivocal changes within endocannabinoid signaling induced by various types of exercise; thus, this topic needs further investigation in future studies.

## 5. The Triad—Physical Activity, the Endocannabinoid System, and the Novel Therapeutic Approach to Metabolic Disorders—How All These Components May Be Linked?

The summary of the correlation between physical activity, its effect on the ECS, and subsequent metabolic pathologies is presented in the Figure 2.

There is no doubt that the involvement of the ECS in the pathogenesis of metabolic pathologies is crucial [105]. Therefore, targeting the elements of ECS, including receptors, endocannabinoids, and enzymes, by physical exercises and a healthy diet may exert robust beneficial outcomes in either prevention or supportive treatment of metabolic disorders [9]. To date, only a few studies conducted on humans (Table 1) and animal models, namely Wistar rats and C57B1/6J mice (Table 2), showed the positive influence of chronic and acute PA (e.g., treadmill running, swimming, and aerobic exercises) or combined with the appropriate caloric restriction on the endocannabinoid signaling and subsequently, auspicious course of the metabolic syndrome, for instance, body weight loss, decreased waist circumference and visceral adipose tissue percentage, improved insulin signaling, or enhanced lipid profile. Additionally, it is noteworthy that physical exercises and accompanying increased concentration of eCBs significantly improved mood, sense of well-being, and increased motivation through reward systems that may constitute important factors for maintaining patient cooperation with a physician, thereby providing efficient patient compliance, which is crucial during therapy targeting metabolic diseases. These effects probably arise from the fact that CB1R shares similar localization in the midbrain with dopamine receptors and may enhance the activity of dopaminergic neurons in ventral tegmentum and substantia nigra [106]. Interestingly, observed decreased blood eCBs levels after PA among obese individuals are opposite to those described in the studies involving healthy patients, which, as was mentioned above, indicated increased concentrations of AEA and/or 2-AG [91,92,93,94,98,99]. A similar concern deals with the altered expression of cannabinoid receptors. Indeed, a lot of experts suggested that PA in healthy animals was correlated with upregulated CB1R expression within brain structures which, on the contrary, was lowered in the brain and subcutaneous/visceral adipose tissue of obese mice. Neither animal nor human studies revealed consistent outcomes concerning the expression of genes encoding both cannabinoid receptors and enzymes involved in eCBs metabolism like FAAH, DAGL, MAGL, and NAPE-PLD (Table 1 and Table 2). However, Schönke et al., using the online tool MetaMEx, found downregulated skeletal muscle expression of DAGLβ as well as catabolic enzymes, namely MAGL and FAAH, in healthy individuals and patients with metabolic syndrome who underwent chronic aerobic or resistant exercises [107]. It is noteworthy that in vivo studies conducted by Gamelin et al. revealed that obese Wistar rats after a 12-week running period showed an increased expression of TRPV1 and CB1R both in the hippocampus or subcutaneous adipose tissue, which suggests potentially enhanced signaling within this ion channel and may result in the favorable course of obesity and other metabolic conditions [10,108]. In conclusion, all discrepancies and concerns related to ECS signaling suggest unclear and paradoxical involvement of the ECS in both physiological and impaired energy balance and might indicate possible compensation mechanisms in response to the significantly diminished metabolic homeostasis. It strengthens the evidence that the ECS itself constitutes a complex molecular apparatus that may be involved in producing beneficial therapeutic effects and may act as a culprit due to confirmed dysregulation in metabolic diseases [107]. A mutual correlation between the ECS and PA should be discussed more in-depth in the future considering the significant deficit of the studies in this area.

## 6. Conclusions

To the best of our knowledge, this is the first review directly and comprehensively discussing the uncharted link between physical activity and its influence on the endocannabinoid signaling in the aspect of beneficial effects in the management of metabolic disorders. Considering the very alarming worldwide prevalence of these diseases as well as the unexplored potential of the topic, we believe that this paper, at least in part, will encourage researchers toward investigating this interesting, yet very complicated interplay. ECS and physical activity constitute robust and valuable therapeutic and preventive approaches that may significantly contribute to the decreased socioeconomic burden and the reduced annual number of patients suffering from obesity and other metabolic disorders. The future investigation should primarily encompass further discovery of the link between physical activity, alterations within endocannabinoid signaling and subsequently improved metabolic status of overweight, obese, and diabetic individuals.

## Figures and Tables

**Figure 1 ijms-23-03083-f001:**
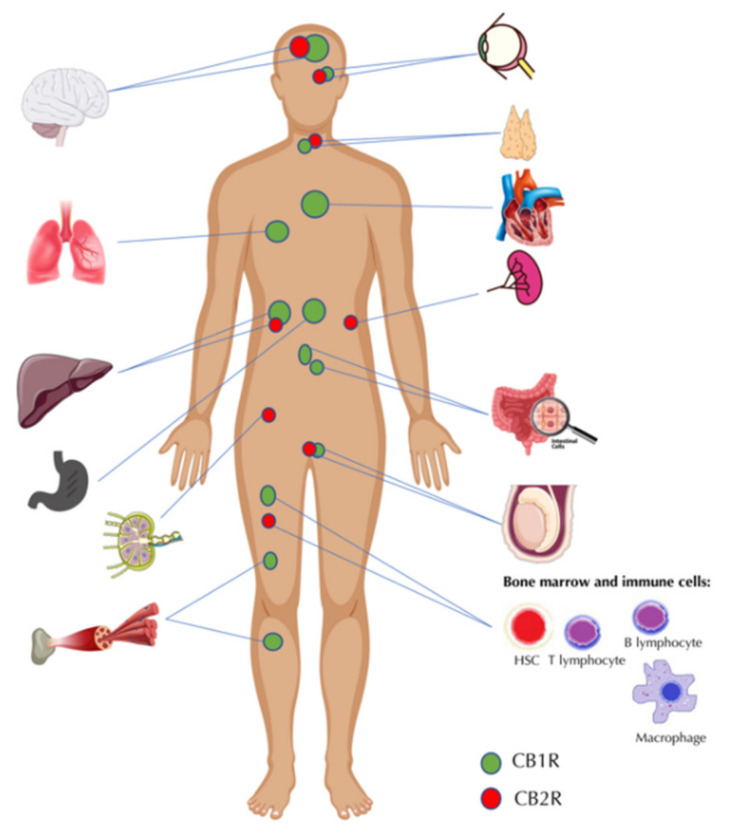
The distribution of cannabinoid receptors within the human body. CB1R—cannabinoid receptor type 1, CB2R—cannabinoid receptor type 2, HSC—hematopoietic stem cell (some graphic was acquired from vecteezy.com, accessed date: 1 November 2021).

**Figure 2 ijms-23-03083-f002:**
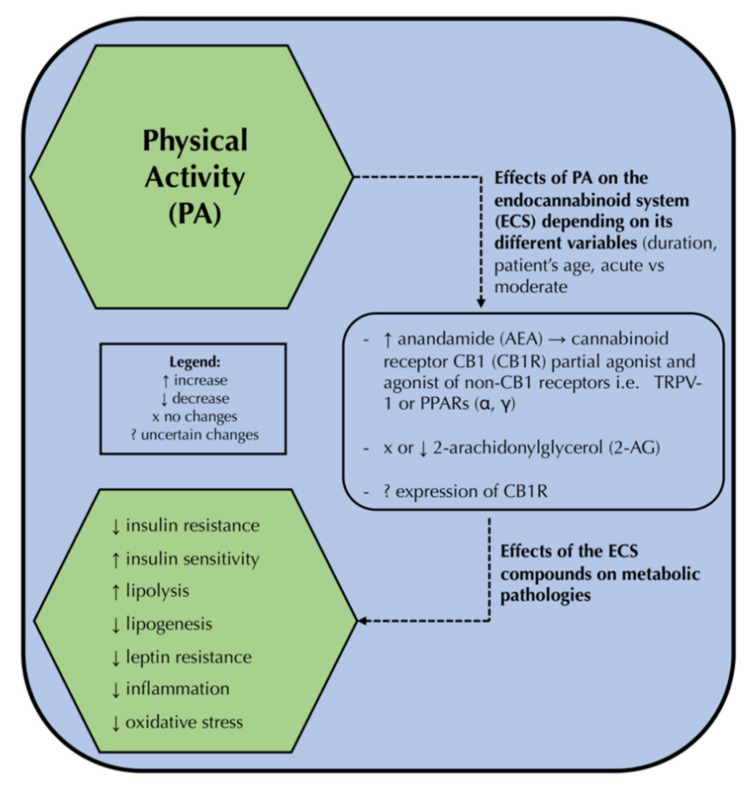
The effect of physical activity on the endocannabinoid system components and, subsequently, its possible impact on the attenuation of metabolic pathologies. PA: physical activity, ECS: endocannabinoid system, AEA: anandamide, CB1R: cannabinoid receptor type 1, ECS: the endocannabinoid system, TRPV-1: transient receptor potential vanilloid 1 ion channel, PPAR: peroxisome proliferator-activated receptors α, 2-AG: 2-arachidonoylglycerol.

**Table 1 ijms-23-03083-t001:** A summary of the clinical studies showing the correlation between physical activity, alterations in the endocannabinoid system, and subsequent beneficial therapeutic effects in metabolic pathologies.

Subjects	Performed Activity	Main Outcomes	Reference
Viscerally obese men (*n* = 49)	1-year lifestyle modification program including regular physical activity and healthy diet	**ECS:** ↓ plasma AEA and 2-AG **Metabolic effects:** ↓ body weight and waist circumference ↓ visceral adipose tissue ↑ HDL_3−_C	[109]
Overweight or obese women (*n* = 30)	20 weeks of moderate (HR_max_ = 45–50%) or intense aerobic exercises (HR_max_ = 70–75%) combined with caloric restriction	**ECS:** ↑ *cb1r* gluteal adipose tissue gene expression ↓ *faah* abdominal adipose tissue gene expression **Metabolic effects:** ↓ body weight and waist circumference ↓ glucose ↓ insulin	[110]
Obese women (*n* = 77)	6 days of normal, daytime physical activity and 6 days of moderate–vigorous physical activity	**ECS:** ↓ plasma 2-AG ↑ plasma AEA and OEA (only for moderate–vigorous physical activity) **Metabolic effects:** ↓ BMI and waist circumference (only for moderate–vigorous physical activity)	[111]

2-AG—2-arachidonylglycerol, AEA—anandamide, BMI—body mass index, *cb1r*—cannabinoid type 1 receptor gene, ECS—endocannabinoid system, *faah*—fatty acid amide hydrolase gene, HDL_3_-C—high density lipoprotein—cholesterol, HR_max_—maximum heart rate, OEA—oleoylethanolamide, ↑—increase, ↓—decrease.

**Table 2 ijms-23-03083-t002:** A summary of the animal studies showing the correlation between physical activity, alterations in the endocannabinoid system, and subsequent beneficial therapeutic effects in metabolic pathologies.

Subjects	Performed Activity	Main Outcomes	Reference
Male Wistar rats fed with HFD	1 h of swimming, 3 times a week for 6 months	**ECS:** ↓ expression of CB1R in VAT and SAT ↑ expression of PPAR*δ* in VAT **Metabolic effects:** ↓ body weight ↓ visceral adipose tissue percentage ↓ blood pressure	[112]
C57Bl/6J male mice fed with HFD	1 h of treadmill running, 6 times a week for 6 weeks	**ECS:** ↓ plasma AEA and 2-AG ↓ CB1R and CB2R expression in the brain ↓ CB2R expression in the epididymal fat ↓ MAGL, DAGL- α and β, FAAH, and NAPE-PLD expression in the brain and epididymal fat **Metabolic effects:** ↓ body weight ↓ body fat percentage ↓ LDL-C ↓ TG ↑ HDL-C	[113]
Male Wistar rats fed with HFD	1 h of treadmill running, 5 times a week for 12 weeks (70–80% MAV)	**ECS:** no effect on AEA and 2-AG in SAT and VAT ↑ DAGL-α and FAAH expression in SAT ↑ *cb1r* and *trpv1* gene expression in SAT **Metabolic effects:** ↓ body weight ↓ fasting plasma glucose	[10]
Male Wistar rats fed with HFD	1 h of treadmill running 5 times a week for 12 weeks (70–80% MAV)	**ECS:** ↑ *cb1r* and *trpv1* gene expression in hippocampus ↑ DAGL-α expression in hippocampus ↑ *faah* gene expression in hippocampus ↓ *napepld* gene expression in hippocampus **Metabolic effects:**↓ body weight ↓ fasting plasma glucose	[108]

2-AG—2-arachidonylglycerol, AEA—anandamide, *cb1r*—cannabinoid receptor type 1 gene, CB1R—cannabinoid receptor type 1, CB2R—cannabinoid receptor type 2, DAGL α and β—diacylglycerol lipase α and β, ECS—the endocannabinoid system, *faah*—fatty acid amide hydrolase gene, FAAH—fatty acid amide hydrolase, HDL-C—high-density lipoprotein cholesterol, HFD—high-fat diet, LDL-C—low-density lipoprotein cholesterol, MAGL—monoacyl glycerol lipase, MAV—maximal aerobic velocity, NAPE-PLD—N-Acyl phosphatidylethanolamine phospholipase D, *napepld*—N-acyl phosphatidylethanolamine phospholipase D gene, PPAR*δ*—peroxisome proliferator activation receptor *δ*, SAT—subcutaneous adipose tissue, TG—triglyceride, *trpv1*—transient receptor potential vanniloid 1 ion channel gene, VAT—visceral adipose tissue, ↑—increase, ↓—decrease.

## Data Availability

Not applicable.

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
