# Peer review of "The Endocannabinoid System and Physical Activity—A Robust Duo in the Novel Therapeutic Approach against Metabolic Disorders"

_ijms, 2022, doi:10.3390/ijms23063083_

Round 1

Reviewer 1 Report

The author reviewed the paper about “The Endocannabinoid System and Physical Activity – a Robust 2 Duo in the Novel Therapeutic Approach Against Metabolic 3 Disorders”.

This is an interesting review paper and I have some minor corrections

  1. Please change the Ca2+ in line 85 with 2+ in superscript.
  2. The author mentioned cannabinoid receptor type 1 in lines 179 and 247. You can give the abbreviation here.
  3. I would suggest changing the color for figure 2 with a big font size.

Author Response

Bialystok, 08/03/2022

Dear Madam, Dear Sir,

We express our sincere thanks for the time and effort that you dedicated to reviewing our manuscript. We are also grateful for the positive remarks on our work. We have been able to incorporate all the concerns related to the submitted manuscript. All the changes within the manuscript have been highlighted.

  1. Please change the Ca2+ in line 85 with 2+ in superscript.

Authors: It was corrected (line: 86) .

  1. The author mentioned cannabinoid receptor type 1 in lines 179 and 247. You can give the abbreviation here.

Authors: It was corrected (lines: 216 and 287).

  1. I would suggest changing the color for figure 2 with a big font size.

Authors: Thank You for your suggestion. We changed the color of figure 2 to be more eye-friendly likewise the font has been magnified.

Considering all the Reviewer’s suggestions and concerns provided, we believe that the manuscript was significantly improved and is now more suitable for publication in the International Journal of Molecular Sciences.

Yours Faithfully,

Hubert Zywno

Department of Physiology,

Medical University of Bialystok, Poland

Reviewer 2 Report

Key note

The authors of the review focused on the relationship between the activity of the endocannabinoid system (ECS) and the body's energy expenditure during physical exertion. A review of existing research data in this area can serve as a rationale for a new approach to the treatment of metabolic disorders, the pathogenesis of which involves disorders of the endocannabinoid system. In reviewing the ECS, the authors focused mainly on the effects of anandamide and 2-arachidonoylglycerol as primary ligands for type 1 and 2 cannabinoid receptors. However, the modern view of ECS also includes endovanilloids, in particular acyl dopamines, which, along with anandamide, are ligands for the thermosensitive and pH-sensitive receptor channel TRPV1. Unfortunately, the authors missed in their review the consideration of the effects mediated by this receptor on the body's energy homeostasis. Authors are encouraged to fill this gap. For example, refer to the article by Christie S, Wittert GA, Li H, Page AJ. Involvement of TRPV1 Channels in Energy Homeostasis. Front Endocrinol (Lausanne). 2018 Jul 31;9:420. doi: 10.3389/fendo.2018.00420. PMID: 30108548. And also, please, include material from other articles on this topic in the review.

Minor remark

The list of cited sources should be brought to a unified form and the bibliographic data for references 30, 33, 34, etc. should be supplemented

Author Response

Bialystok 08/03/2022

Dear Madam, Dear Sir,

We express our sincere thanks for the time and effort that you dedicated to reviewing our manuscript. We have been able to incorporate all the concerns related to the submitted manuscript. All the changes within the manuscript have been highlighted.

Key note

The authors of the review focused on the relationship between the activity of the endocannabinoid system (ECS) and the body's energy expenditure during physical exertion. A review of existing research data in this area can serve as a rationale for a new approach to the treatment of metabolic disorders, the pathogenesis of which involves disorders of the endocannabinoid system. In reviewing the ECS, the authors focused mainly on the effects of anandamide and 2-arachidonoylglycerol as primary ligands for type 1 and 2 cannabinoid receptors. However, the modern view of ECS also includes endovanilloids, in particular acyl dopamines, which, along with anandamide, are ligands for the thermosensitive and pH-sensitive receptor channel TRPV1. Unfortunately, the authors missed in their review the consideration of the effects mediated by this receptor on the body's energy homeostasis. Authors are encouraged to fill this gap. For example, refer to the article by Christie S, Wittert GA, Li H, Page AJ. Involvement of TRPV1 Channels in Energy Homeostasis. Front Endocrinol (Lausanne). 2018 Jul 31;9:420. doi: 10.3389/fendo.2018.00420. PMID: 30108548. And also, please, include material from other articles on this topic in the review.

Authors: Thank you for your valuable advice. We agree that the involvement of ion channel TRPV1 in endocannabinoid signaling is very important and should be implemented in our review. Detailed information about the effects of TRPV1 on the body’s energy homeostasis likewise the interplay with ECS has been added (lines:144-163). Furthermore, we decided to describe the link between altered expression of this channel and deteriorated ECS signaling during impaired metabolic balance in obesity (lines:203-208) as well as we described the results of in vivo studies conducted by Gamelin et al. showing the increased expression of TRPV1 and CB1R both in subcutaneous tissue and hippocampus which was induced by physical activity in Wistar rats [1,2] (lines:406-411). Those studies allowed us to underline the influence of physical activity in metabolic disorders on the signaling within ECS and TRPV1. We referred to the provided by the Reviewer article as well as other references have been added.

Minor remark

The list of cited sources should be brought to a unified form and the bibliographic data for references 30, 33, 34, etc. should be supplemented

Authors: Thank you for pointing this out. All the references have been scrupulously checked and revised to the united form and the bibliographic data have been supplemented where was necessary as was suggested.

References:

[1]. Gamelin FX, Aucouturier J, Iannotti FA, Piscitelli F, Mazzarella E, Aveta T, Leriche M, Dupont E, Cieniewski-Bernard C, Montel V, Bastide B, Di Marzo V, Heyman E. Effects of chronic exercise on the endocannabinoid system in Wistar rats with high-fat diet-induced obesity. J Physiol Biochem. 2016 Jun;72(2):183-99. doi: 10.1007/s13105-016-0469-5. Epub 2016 Feb 15. PMID: 26880264.

[2]. Gamelin FX, Aucouturier J, Iannotti FA, Piscitelli F, Mazzarella E, Aveta T, Leriche M, Dupont E, Cieniewski-Bernard C, Leclair E, Bastide B, Di Marzo V, Heyman E. Exercise training and high-fat diet elicit endocannabinoid system modifications in the rat hypothalamus and hippocampus. J Physiol Biochem. 2016 Aug;73(3):335-347. doi: 10.1007/s13105-017-0557-1. Epub 2017 Mar 10. Erratum in: J Physiol Biochem. 2017 Oct 5;: PMID: 28283967.

Considering all the Reviewer’s suggestions and concerns provided, we believe that the manuscript was significantly improved and is now more suitable for publication in the International Journal of Molecular Sciences.

Yours Faithfully,

Hubert Zywno

Department of Physiology,

Medical University of Bialystok, Poland
